# Does the Timing of Response Impact the Outcome of Relapsed/Refractory Acute Myeloid Leukemia Treated with Venetoclax in Combination with Hypomethylating Agents? A Proof of Concept from a Monocentric Observational Study

**DOI:** 10.3390/jcm14155586

**Published:** 2025-08-07

**Authors:** Ermelinda Longo, Fanny Erika Palumbo, Andrea Duminuco, Laura Longo, Daniela Cristina Vitale, Serena Brancati, Cinzia Maugeri, Marina Silvia Parisi, Giuseppe Alberto Palumbo, Giovanni Luca Romano, Filippo Drago, Francesco Di Raimondo, Lucia Gozzo, Calogero Vetro

**Affiliations:** 1Hematology and Bone Marrow Transplantation Unit, A.O.U. Policlinico “G.Rodolico—S. Marco”, 95123 Catania, Italy; lindalongo96@gmail.com (E.L.); fannypalumbo@gmail.com (F.E.P.); maugericinzia@hotmail.com (C.M.); marinaparisi@hotmail.it (M.S.P.); palumbo.ga@gmail.com (G.A.P.); francesco.diraimondo@unict.it (F.D.R.); 2Clinical Pharmacology Program/Regional Pharmacovigilance Centre, A.O.U. Policlinico “G.Rodolico—S. Marco”, 95123 Catania, Italy; longo@policlinico.unict.it (L.L.); danielac.vitale@gmail.com (D.C.V.); serena.brancati@gmail.com (S.B.); f.drago@unict.it (F.D.); luciagozzo86@icloud.com (L.G.); 3Department of Medical, Surgical Sciences and Advanced Technologies “G.F. Ingrassia”, University of Catania, 95123 Catania, Italy; 4Department of Medicine and Surgery, University of Enna “Kore”, 94100 Enna, Italy; giovanniluca.romano@unict.it; 5Hematology and Bone Marrow Transplantation Unit, Hospital of Bolzano (SABES-Azienda Sanitaria dell’Alto Adige), Teaching Hospital of Paracelsus Medical University, 39100 Bolzano, Italy; gerovetro@gmail.com

**Keywords:** Acute Myeloid Leukemia, cytopenia, early response, hypomethylating Agents, late response, relapsed/refractory AML, response rate, treatment related-toxicity, venetoclax, washout period

## Abstract

**Background**: Relapsed/refractory acute myeloid leukemia (R/R AML) remains a therapeutic challenge due to disease heterogeneity, resistance mechanisms, and poor tolerability to intensive regimens. Venetoclax (VEN), a BCL-2 inhibitor, has shown promise in combination with hypomethylating agents (HMAs), but data on response timing in the R/R setting are limited. The aim of this study was to assess the efficacy, safety, and kinetics of response to HMA-VEN therapy in a real-world cohort of R/R AML patients, with particular focus on early versus late responders. **Methods**: This prospective single-center study included 33 adult patients with R/R AML treated with VEN plus either azacitidine (AZA) or decitabine (DEC) from 2018 to 2021. The primary endpoint was the composite complete remission (cCR) rate and the rate of early and late response, respectively, occurring within two cycles of therapy or later; secondary endpoints included overall survival (OS), relapse-free survival (RFS), time to relapse (TTR), and safety. **Results**: The cCR was 58%, with complete remission (CR) or CR with incomplete recovery (CRi) achieved in 52% of patients. Median OS was 9 months. No significant differences in OS or TTR were observed between early (≤2 cycles) and late (>2 cycles) responders. Eight responders (42%) underwent allogeneic hematopoietic stem cell transplantation (HSCT), with comparable transplant rates in both groups of responders. Toxicity was manageable. Grade 3–4 neutropenia occurred in all patients, and febrile neutropenia occurred in 44% of patients. An Eastern Cooperative Oncology Group (ECOG) score >2 was associated with inferior response and shorter treatment duration. **Conclusions**: HMA-VEN therapy is effective and safe in R/R AML, including for patients with delayed responses. The absence of a prognostic disadvantage for late responders supports flexible treatment schedules and suggests that the continuation of therapy may be beneficial even without early blast clearance. Tailored approaches based on performance status and comorbidities are warranted, and future studies should incorporate minimal residual disease (MRD)-based monitoring to refine response assessment.

## 1. Introduction

Acute myeloid leukemia (AML) is a clonal hematopoietic stem cell disorder characterized by the accumulation of immature myeloid precursors in the bone marrow and peripheral blood. These leukemic blasts impair normal hematopoiesis and lead to progressive bone marrow failure, manifesting clinically as anemia, infections, and hemorrhagic events [1,2]. Despite advancements in diagnostics and supportive care, prognosis remains poor, particularly for elderly or relapsed/refractory (R/R) patients. Standard induction chemotherapy, typically a combination of cytarabine and anthracycline (“7 + 3” regimen), can induce complete remission in 60–80% of younger patients [3,4]. However, its use in elderly individuals is limited by comorbidities and reduced physiological reserve [5]. For these patients, less intensive regimens, such as low-dose cytarabine (LDAC) or hypomethylating agents (HMAs), including azacitidine (AZA) and decitabine (DEC), have been used, although with modest survival benefits [6,7,8,9,10]. The introduction of targeted therapies has reshaped AML treatment. Venetoclax (VEN), an oral selective inhibitor of BCL-2, represents one such breakthrough. BCL-2 inhibition restores the intrinsic apoptosis pathway by releasing pro-apoptotic proteins like Bcl-2-associated X protein (BAX) and Bcl-2 homologous antagonist/killer (BAK) [11,12,13,14,15]. VEN thus induces apoptosis in leukemic cells, especially when used in combination with agents that downregulate other anti-apoptotic proteins, such as HMAs [13,16].

Initially developed for chronic lymphocytic leukemia, VEN has shown significant efficacy in AML, particularly when combined with AZA or DEC. The VIALE-A phase III trial demonstrated that VEN combined with AZA improved overall survival (OS) and response rates in treatment-naïve AML patients unfit for intensive chemotherapy compared to AZA alone, leading to its regulatory approval for this indication and worldwide use [7,8,9,17,18,19,20,21,22]. However, evidence for VEN in R/R AML remains less robust and is largely derived from retrospective analyses and off-label use [10,23,24].

Patients with R/R AML face limited treatment options due to cumulative toxicity, refractory disease biology, and poor performance status, with a median OS of less than six months [25,26]. Consequently, there is a pressing need for effective, well-tolerated salvage regimens. VEN-based regimens have gained interest in the R/R setting, and recent real-world studies suggest that HMA-VEN combinations may achieve response rates of 30–60% in R/R AML patients, particularly when administered earlier in the disease course or in patients with favorable genetics [25,26,27,28,29,30]. However, treatment outcomes may vary substantially depending on prior therapies, patient-related factors, and the specific treatment schedule. Although VEN-based regimens have shown the ability to induce rapid and deep remissions in AML, the prognostic significance of early versus late hematologic responses remains insufficiently characterized, particularly in the R/R setting. A post hoc analysis from the VIALE-A trial showed that most responses to AZA-VEN occurred during the first treatment cycle (i.e., 76% of responders), leading to recommendations for early bone marrow assessment. Yet, whether the timing of response has prognostic or therapeutic implications in the R/R setting is still unclear.

In our monocentric cohort, we specifically investigated the kinetics of response to HMA-VEN. In this context, we conducted a study to evaluate the clinical efficacy and safety of VEN in combination with either AZA or DEC in adult patients with R/R AML treated at the University Hospital of Catania. Our aim was to provide additional real-world evidence on the role of HMA-VEN therapy in this difficult-to-treat population.

## 2. Materials and Methods

### 2.1. Study Design and Setting

This was a prospective, single-center, observational pharmacovigilance study conducted at the Hematology Unit of the University Hospital of Catania, Italy, focused principally on the monitoring of adverse events of off-label drugs. The study was reviewed and approved by the Institutional Review Board Catania 1, file number 37/2017/PO, on 13 February 2017.

This subanalysis evaluated the efficacy and safety of VEN in combination with HMAs, either AZA or DEC, in 33 adult patients with R/R AML. The observation period spanned from January 2018 to December 2021. All procedures were performed in accordance with the Declaration of Helsinki, and patient consent for data analysis was obtained when appropriate. In total, sixteen out of 33 patients were also enrolled in the multicenter observational AVALON study, further validating the consistency and representativeness of our cohort within a broader national context [10,24,31,32].

### 2.2. Patient Population

Eligible patients were adults (≥18 years old) with a confirmed diagnosis of R/R AML according to the World Health Organization (WHO) 2016 classification, treated with the combination HMA-VEN [33]. R/R AML was defined as failure to achieve complete remission (CR) following induction therapy (refractory, REF) or as hematologic relapse after a previously achieved CR (relapsed, REL). REL patients were further subdivided into early and late relapse, based on the timing of disease recurrence after prior therapy: early relapse was defined as <12 months, while late relapse was defined as ≥12 months.

The inclusion criteria were as follows:Diagnosis of AML;Patients must receive VEN in combination with AZA or DEC as an off-label treatment strategy;Second or subsequent lines of therapy.

The exclusion criterion was
Patients with prior exposure to VEN in other settings were excluded.

Data were extracted from the electronic medical records and anonymized for analysis. Performance status was assessed using the Eastern Cooperative Oncology Group (ECOG) scale [34].

### 2.3. Treatment Regimens

VEN was administered orally in a dose-escalation scheme during the first cycle: 100 mg on day 1, 200 mg on day 2, and 400 mg from day 3 onward, with food. From the second cycle, VEN was continued at a fixed dose of 400 mg/day, adjusted for concomitant use of CYP3A4 inhibitors such as azole antifungals [35]. AZA was administered subcutaneously at a dose of 75 mg/m^2^ on days 1–7 of each 28-day cycle. DEC was administered intravenously at 20 mg/m^2^ on days 1–5. The treating physician chose HMA (AZA or DEC). Supportive care, including transfusions and prophylactic antimicrobials, was provided according to institutional protocols.

Treatment interruption criteria were defined as follows: VEN was withheld in the event of grade 4 neutropenia (ANC < 500/μL) or thrombocytopenia (platelets < 25,000/μL). Therapy was resumed only after adequate hematologic recovery (ANC ≥ 500/μL and platelets ≥ 25,000/μL), and venetoclax was reintroduced at a reduced duration or dose (e.g., from 28 days to 21 or 14 days per cycle) [36,37,38,39]. Febrile neutropenia (FN) was defined as a single oral temperature greater than or equal to 38.3 °C or a temperature greater than or equal to 38 °C for at least an hour, with an absolute neutrophilic count of less than 1500 cells/μL [40]. Invasive fungal infections were defined according to the European Organization for Research and Treatment of Cancer/Mycoses Study Group (EORTC/MSG) Consensus Criteria [41].

### 2.4. Molecular and Cytogenetic Evaluation

Cytogenetic analysis was performed using G-banding techniques, and cytogenetic risk was assessed according to Medical Research Council (MRC) criteria [42], and molecular profiling was conducted to detect mutations in the genes *NPM1* and *FLT3* for all patients [43,44]. Due to resource and technical constraints at the time of data collection—conducted in a routine clinical setting and not within a prospective genomic screening protocol—and given the lack of validation of the 2017 European LeukemiaNet (ELN) risk stratification in R/R AML patients treated with HMA-VEN, additional mutational data (such as *TP53*, *ASXL1*, and *RUNX1*) were not available.

### 2.5. Response Criteria

Response to treatment was evaluated according to ELN 2017 criteria [45]: complete remission (CR): <5% blasts in bone marrow, ANC > 1000/μL, and platelets > 100,000/μL; CR with incomplete hematologic recovery (CRi): <5% blasts with ANC < 1000/μL or platelets < 100,000/μL; morphologic leukemia-free state (MLFS): <5% marrow blasts, and no hematologic recovery required; partial response (PR): reduction in marrow blasts by ≥50% to 5–25%; progressive disease (PD): increase in bone marrow or peripheral blood blasts; stable disease (SD): not meeting criteria for response or progression; and relapse: bone marrow blasts ≥5%, or reappearance of blasts in the blood, or development of extramedullary disease.

Patients were considered early responders if CR/CRi/MLFS were reached after one to two cycles, whereas others were considered late responders [46].

Primary refractory disease was defined as a lack of reduction of bone marrow (BM) blasts to 5% or less by up to cycle 4 of VEN+HMA [47].

### 2.6. Endpoints

The primary endpoint was the composite complete remission (cCR) rate, defined as the sum of CR, CRi, and MLFS. Due to the observational nature of this study, partial remission (PR) was not considered a response [45].

Secondary endpoints included OS, relapse-free survival (RFS), time to relapse (TTR), and safety [48,49,50].

OS was defined as the time from the initiation of HMA-VEN therapy to death from any cause. RFS was defined as the time from achievement of response (either CR or CRi or MLFS) to disease progression or death. TTR was measured from the date of achievement of remission until the date of hematologic relapse, excluding deaths from other causes, and was used to describe remission duration.

### 2.7. Safety Assessment

Adverse events were graded according to the National Cancer Institute Common Terminology Criteria for Adverse Events (CTCAE) version 5.0. Hematologic toxicities, infectious complications (including febrile neutropenia), and hepatic, gastrointestinal, and dermatologic events were monitored throughout treatment [51].

### 2.8. Statistical Analysis

Descriptive statistics were used to summarize baseline patient characteristics, treatment regimens, and adverse events. Continuous variables were reported as medians and ranges or means ± standard deviation (SD). Categorical variables were expressed as frequencies and percentages. The chi-square or Fisher’s exact test was used to compare categorical variables. The distribution of continuous variables was assessed using the Shapiro–Wilk test. Variables with a normal distribution were expressed as mean ± standard deviation and compared using the Student’s *t*-test. Non-normally distributed variables were expressed as median and compared using the Mann–Whitney U test. Survival analyses were performed using the Kaplan–Meier method. The log-rank test was used to compare survival distributions between groups. Statistical significance was defined as *p* < 0.05. All analyses were conducted using SPSS version 20 (IBM Corp., Armonk, NY, USA).

## 3. Results

### 3.1. Patient Characteristics

A total of 33 adult patients with R/R AML were treated with VEN in combination with HMA from 2018. The median age was 58 years (range 23–79), with a nearly equal sex distribution (16 males, 17 females). At baseline, 15 patients (45%) were deemed unfit for intensive chemotherapy due to age, comorbidities, or PS, or previous treatment-related toxicities, and 18 (55%) had no other therapeutic alternatives. According to the ECOG scale, 28 patients had a PS of 0–2, and five patients had a PS of 3–4.

Regarding previous treatments, 19 out of the 33 enrolled patients (57.5%) received standard 7 + 3 induction therapy. Among them, four were refractory: three proceeded directly to AZA/VEN, while one received salvage chemotherapy with FLA-Ida, a high-dose cytarabine plus midostaurin, followed by AZA, but proved resistant to all treatments. The remaining 15 patients experienced disease relapse, 10 of whom relapsed early. Among these, four were treated directly with HMA/VEN, and two initially received AZA with clinical response but later relapsed and were switched to AZA/VEN (one of them also received gilteritinib without clinical benefit). Three patients underwent salvage therapy with FLA-Ida (fludarabine, high-dose cytarabine, and idarubicin) and showed a transient response but relapsed early and were subsequently treated directly with HMA/VEN, except for two who first received AZA monotherapy and again experienced early relapse. One additional patient relapsed early following allogeneic HSCT. Five patients experienced late relapse. Two of them were classified as low-risk at diagnosis and had achieved minimal residual disease (MRD) negativity after induction and consolidation therapy. Due to age >70 years, two patients were maintained on AZA during follow-up. One additional patient experienced late relapse after HSCT. Out of a total of 33, 2 patients (6.1%) were refractory to induction therapy with CPX-351 and were subsequently treated with AZA/VEN. In total, among patients previously exposed to intensive chemotherapy, seven (36.8%) received AZA before AZA/VEN combination. The remaining twelve patients (36.4%) received HMA alone as first-line therapy (four DEC, 12.1%, and eight AZA, 24.2%), with VEN added upon resistance or relapse.

At the end, 14 patients were never exposed to HMA before HMA-VEN initiation, 7 patients initially treated with intensive chemotherapy underwent AZA before AZA-VEN, and 12 patients were previously treated with sole HMA.

Among the 33 patients, 26 (79%) received HMA-VEN therapy in second-line treatment, while 7 patients (21%) received it in third-line treatment or beyond. Of these, three patients (9%) had relapsed after an allogeneic hematopoietic stem cell transplantation (allo-HSCT). Four patients (12%) had therapy-related AML (t-AML), and ten patients (30%) had AML with myelodysplasia-related changes (AML-MRC). Genetic profiling at baseline revealed that seven patients (21%) harbored *NPM1* mutations, and four (12%) had *FLT3*-ITD mutations. Adverse cytogenetic features, including complex karyotype and monosomy 5 or 7, and del(5q)/del(7q) were observed in nine patients (27%).

The reasons why patients were referred to HMA/VEN treatment without pursuing other chemotherapies are summarized in Table 1. Table 2 summarizes patients’ features at baseline.

The pre-treatment median WBC was 3120/uL (range 780–52,660/uL). The median pre-treatment bone marrow blast was 30%, ranging between 12 and 100%.

### 3.2. Therapies Administred

VEN was administered according to a 3-day ramp-up protocol and then maintained at 400 mg/day, with adjustments based on antifungal prophylaxis (AFP). Most patients (85%) received AZA as the HMA backbone, while five patients (15%) received DEC. Four of the five patients treated with DEC had previously received DEC monotherapy, and the remaining one had relapsed after allo-HSCT. The treatment was generally well tolerated, with dose interruptions required in a subset of patients due to cytopenia or infectious complications, as previously published [8].

AFP was administered as follows: the majority received fluconazole (n = 23, 71.9%), followed by posaconazole (n = 7, 21.9%), and caspofungin (n = 1, 3.1%). In two patients (6.3%), no AFP was administered during treatment.

All patients received antiviral prophylaxis with acyclovir. No antibiotic prophylaxis was administered during the treatment period.

### 3.3. Composite Complete Remission (cCR) Rate

Of the 33 patients treated, 19 responded to therapy, resulting in a cCR of 58%. Specifically, 17 patients (52%) achieved CR, and 2 patients (6%) achieved an MLFS (Table 2). Twelve patients were early responders (after cycles 1–2), while seven responded later (cycles 3–4). Differences between early and late responders are shown in Table 3.

Treatment duration varied, with a median of three cycles (range: 1–14). Notably, one patient completed 14 cycles, one patient received 11 cycles, and another patient completed 10 cycles. One patient discontinued after five cycles. Three patients each received four cycles, and another three patients completed three cycles.

The majority of remaining responders underwent shorter treatment courses: six patients received only two cycles, and three discontinued after a single cycle.

Three patients (9%) achieved PR, one of them died after cycle 2 due to febrile neutropenia (FN), while the remaining two patients had persisting PR after cycle 4. Nine (27%) patients were refractory to therapy after cycle 4, while two patients (6%) died during the first cycle, one due to sepsis and the other from COVID-19 complications and fungal superinfection. Among the 14 non-responding patients, including the two who died from infectious complications, the median number of administered cycles was two (range 1–9), with a mean of approximately 3.2 cycles. In addition to the two patients who died during the first cycle, three patients were declared resistant after receiving only one cycle and were unable to continue treatment due to worsening clinical conditions. Two patients surprisingly completed up to nine cycles, but both showed persistent stable disease with bone marrow blast percentages consistently ranging between 20% and 30%.

The number of treatment cycles did not significantly differ between responders and non-responders (Mann–Whitney U test, *p* = 0.144).

Patients treated with AZA-VEN tended to have a higher rate of cCR compared to DEC-VEN, i.e., 64.3% for AZA-VEN vs. 20% for DEC-VEN, *p* = 0.065. From a genetic perspective, mutations in *NPM1* or *FLT3*-ITD, as well as harboring a complex karyotype, did not relate to the cCR rate. PS significantly influenced treatment outcomes. Patients were divided into two groups based on their PS: 0–2 (28 patients) and 3–4 (5 patients). Nineteen out of twenty-eight (67.9%) patients with a 0–2 score responded to therapy, while none of the five patients with a score of 3–4 responded (*p* = 0.005). Responders were also younger than refractory patients (mean age 53 years ± 14.6 SD for responders vs. 63 years ± 8.9 SD for non-responders, *p* = 0.02). The mean WBC count before the start of HMA-VEN did not differ between responders and non-responders. The line of therapy did not relate to the response: 17 of 26 patients (65.4%) receiving HMA-VEN in second-line therapy responded, compared to 2 of 7 patients (28.6%) treated in third-line or beyond, but the difference was not statistically significant (*p* = 0.08). Previous HSCT did not influence the cCR rate.

### 3.4. Survival Outcomes

After a median follow-up of 20 months (95% CI 16–24), median OS for the entire cohort was 9 months (95% CI: 3–14), with nine patients alive at time of analysis. Patients who achieved CR or CRi had significantly better survival (median OS: 15.9 months, 95% CI 12–19) compared to non-responders (median OS: 5 months, 95% CI 2.7–7) (Figure 1). Median RFS was 11.8 months (95% CI: 8.5–15) and median TTR was 15.9 months (95% CI: 10–21). This discrepancy is explained by the fact that three patients died after undergoing transplantation without experiencing a relapse. RFS, TTR, and OS were not influenced by gender, line of treatment (second-line vs. third-line or beyond), therapy association (AZA or DEC), previous HSCT or HMA exposure, nor *NPM1*/*FLT3*-ITD mutations or the occurrence of complex karyotype (Appendix A).

No differences emerged evaluating early vs. late responders in terms of TTR (15.6 months 95% CI 5–26 vs. 15.9, 95% CI 11–20.3, *p* = 0.8, respectively) and OS (17.9 months 95% CI 9–26 vs. 15.6 95%, CI 10–20.7, *p* = 0.4, Figure 2).

Moreover, stratifying patients according to previous treatments, no differences emerged between groups in terms of OS, RFS, and TTR.

Poor performance status (PS) was associated with a trend toward inferior OS, though the difference did not reach statistical significance (PS 0–2: median OS 11.1 months, 95% CI 3.9–18.0 vs. PS 3–4: 7.0 months, 95% CI 0.5–13.0; *p* = 0.057). Only response to treatment was related to OS, as aforementioned.

It is noteworthy that 3 out of 14 non-responding patients (21.4%) acquired *FLT3*-ITD mutation at re-evaluation, while 1 responding patient acquired a *FLT3*-tyrosine kinase domain (TKD) point mutation.

### 3.5. Post-Treatment Transplantation

In total, 8 out of 19 responders (42.1%) underwent allo-HSCT after achieving remission with AZA-VEN therapy. Five patients were bridged directly without further cycles, two after the first cycle and three after two cycles. Three patients were late responders after cycle 3 and proceeded to allo-HSCT after receiving one further post-remission cycle.

The transplant rate was comparable between early and late responders, with 41.7% (5 out of 12) and 42.9% (3 out of 7) of patients undergoing allogeneic HSCT, respectively.

Among transplanted patients, three experienced relapse, one at 4 months and two at 11 months post-transplant, and subsequently died due to disease progression. Five patients initially achieved and maintained remission; however, two died from infections at 9 and 5 months post-transplant, and one died from graft-versus-host disease (GVHD) at 14 months. At the last follow-up, two patients remained alive, in sustained remission and without any signs of GVHD. One patient was scheduled for transplant but experienced disease progression and died of cerebral hemorrhage shortly before conditioning could begin.

### 3.6. Safety Profile

The most common adverse events were hematologic. Grade 3–4 neutropenia occurred in all patients, while 44% experienced febrile neutropenia. Infections—bacterial, fungal, and viral—were managed with antimicrobial prophylaxis and treatment as needed. Non-hematologic toxicities were infrequent: one hepatic toxicity, two cutaneous reactions, and one vestibular disorder were reported. There were no cases of tumor lysis syndrome. Dose interruptions were necessary in 10 patients (30%) due to myelosuppression. VEN was temporarily withheld, and, after hematologic recovery, therapy was resumed at a reduced dose and reduced time-exposure [52].

Among responders, hematological toxicity with FN occurred in eight cases (42.1%).

Hematologic toxicity leading to treatment interruption was significantly more common among late responders (5/7, 71.4%) compared to early responders (3/12, 25.0%; *p* = 0.048). Among early responders, washout periods occurred after the first cycle and ranged between 7 and 15 days. In late responders, treatment holds generally ranged from 7 to 14 days, although one patient required an extended interruption of up to three weeks.

Five out of the seven late responders experienced grade 3–4 infections, compared to three out of the twelve early responders. Although the rate was higher among late responders (57.1% vs. 25.0%), the difference did not reach statistical significance (Fisher’s exact test, *p* = 0.33).

Among late responders, documented infectious events included Ochrobactrum anthropi bacteremia, febrile episodes in a patient colonized with *E. coli* and *Stenotrophomonas maltophilia*, and HHV6 reactivation. One case of fever resolved with empirical meropenem despite persistently negative cultures, while another patient experienced a febrile episode of unknown origin (FUO). Among early responders, polymicrobial infection caused by *Klebsiella Pneumoniae*, *Pseudomonas Aeruginosa*, and *Serratia Marcescens* was reported in one case. Two additional patients developed FUO. Among refractory patients, four cases were classified as fever of unknown origin (FUO) with clinical features consistent with possible invasive fungal infection (IFI), including one patient colonized by a carbapenemase-producing *Klebsiella pneumoniae* (KPC). Notably, all four patients had received fluconazole prophylaxis. One patient developed a soft tissue infection. Two patients had SARS-CoV-2 infection; in one case, this was complicated by a pulmonary superinfection consistent with possible IFI, despite fluconazole prophylaxis.

Patients with infections (n = 16) showed a shorter median OS compared to those without toxicity (n = 17), although the difference was not statistically significant (8.07 vs. 14.53 months; *p* = 0.115).

## 4. Discussion

### 4.1. Clinical Efficacy and Response Kinetics

The treatment of R/R AML remains a formidable challenge due to disease heterogeneity, clonal evolution, and poor responses to traditional salvage regimens [26,53,54,55]. In recent years, the combination of VEN with HMAs, such as AZA or DEC, has garnered growing interest, particularly in older or unfit patients [23,56]. Our real-world data further support this approach: in our single-center retrospective cohort of 33 R/R AML patients, we observed a cCR of 58% and a median OS of 9 months, consistent with outcomes reported in larger multicenter studies [57,58,59].

Notably, although most responses occurred within the first two treatment cycles (63%), a subset of patients exhibited delayed hematologic response. Crucially, no significant differences in OS or TTR were observed between early and late responders. To our knowledge, this is the first study to specifically explore the prognostic impact of response kinetics in the R/R setting. These findings align with observations from the VIALE-A trial, where Pratz et al. demonstrated that the timing of response to VEN-AZA did not predict survival in newly diagnosed AML patients [60]. Our results extend this evidence to the R/R population, suggesting that a delayed response to HMA-VEN therapy does not portend worse outcomes, provided that treatment is continued and adverse events are appropriately managed. This supports a more flexible, individualized approach to treatment duration, even in the absence of early blast clearance.

### 4.2. Hematologic Toxicity and Transplant Outcomes

Interestingly, late responders in our cohort experienced a higher incidence of hematologic toxicity requiring treatment interruption, likely reflecting prolonged marrow suppression in the absence of early leukemic clearance. These patients may undergo sustained cytotoxic pressure in a compromised hematopoietic environment, resulting in delayed recovery. Still, hematologic toxicity did not preclude further treatment or transplant: the rate of FN (44%) was comparable to that reported in other real-world studies [59], and transplant rates were similar between early and late responders [60,61].

Interestingly, the number of cycles between responders and non-responder patients did not differ significantly. This likely reflects the clinical strategy of promptly bridging responding patients to allogeneic HSCT when feasible rather than continuing additional cycles of HMA-VEN therapy. In our cohort, eight responders proceeded to allogeneic HSCT, and five remained in remission at last follow-up. Indeed, the transplant rate did not differ between early and late responders, being approximately 42% for both groups. Transplant bridging remains a pivotal consideration in management, not only in patients newly diagnosed with [61] R/R AML, representing one of the few therapeutic strategies with the potential to significantly improve long-term survival [62]. These outcomes reflect those of Xu et al., who reported frequent transplant bridging in young adults following HMA-VEN salvage therapy [59], and those of the AVALON study, which showed a 20–32% post-response transplant rate [24]. 

### 4.3. Impact of Comorbidities and Performance Status

Importantly, our data support findings by Marconi et al. and others that comorbidity burden and performance status (PS) are critical determinants of HMA-VEN efficacy and tolerability [10]. Patients with lower comorbidity scores experienced significantly longer OS and lower rates of severe non-hematologic adverse events. These findings suggest that comorbidity indices not only reflect biological fitness but also predict treatment tolerability and effectiveness. Importantly, patients with fewer comorbidities were considered more suitable for treatment intensification, including HSCT. Our cohort further supports this association, as patients with ECOG > 2 showed markedly lower cCR and shorter treatment durations and survival compared to those with ECOG 0–2 [63,64]. This suggests that frailty and poor PS limit the ability to achieve remission primarily due to early treatment-related toxicities, which often lead to therapy discontinuation before clinical benefit. However, the observation that some patients achieved remission after multiple cycles highlights the potential for delayed response in this setting. These findings support the need for treatment schedule adaptations, such as reduced venetoclax dose intensity, in frail patients, in order to improve tolerability and allow sufficient treatment exposure to achieve a meaningful response. These results highlight the necessity of integrating comorbidity screening into routine clinical decision-making to optimize outcomes and personalize therapeutic approaches in order to minimize the occurrence of adverse events. 

### 4.4. Infectious Complications and Supportive Care Strategies

Indeed, infectious complications represent a major concern in AML patients receiving VEN-based regimens. Real-world data on infections during HMA-VEN therapy in AML are heterogeneous but overall suggest frequent febrile neutropenia and infections, especially early in treatment, while IFI rates showed a wide range (6 to 26%), with limited benefit from universal antifungal prophylaxis outside high-risk subgroups [65,66,67,68,69]. In our cohort, the incidence of grade 3–4 infections was higher among late responders compared to early responders (57% vs. 25%), although the difference did not reach statistical significance. Importantly, bloodstream infections due to Gram-negative pathogens, viral reactivations, and FUO were observed in both subgroups, confirming the high vulnerability of this population. These findings are in line with previous reports. In a large multicenter study by Candoni et al., infectious complications were documented in over 50% AML patients treated with HMA-VEN, with pneumonia and bloodstream infections being the most frequent events [70]. Moreover, early infections, defined as occurring within the first treatment cycle, were independently associated with inferior OS, as demonstrated by Cordella et al. in a cohort of 263 patients, where early infection and secondary AML emerged as the only independent predictors of poor prognosis [71]. These findings support the hypothesis that infectious events are not merely complications but may also reflect a patient’s intrinsic frailty or disease aggressiveness. Additionally, in patients with chronic lymphocytic leukemia, a predictive score has been proposed to estimate infectious risk prior to initiating VEN therapy based on simple clinical and laboratory parameters [72]. While such tools have not yet been validated in the AML setting, their development could be of value in identifying high-risk individuals who may benefit from preemptive or intensified prophylaxis strategies. Taken together, these data highlight the need for rigorous infectious risk assessment and tailored supportive care during VEN-based therapy, particularly in the early phase of treatment. Future studies are warranted to validate predictive models and integrate them into clinical decision-making in both newly diagnosed and R/R AML settings. 

### 4.5. Antifungal Prophylaxis: Current Evidence and Future Directions

IFI prevention and management remain key challenges, particularly in the context of novel therapies such as VEN and cellular therapies, which are associated with an increased risk of IFI due to their cytotoxic effects. [73,74]. Azole antifungals, commonly used for prophylaxis, can reduce IFI incidence but also significantly increase VEN plasma levels through CYP3A4 inhibition, requiring dose reductions and close toxicity monitoring. Pharmacokinetic and safety data also explored this aspect. A preliminary analysis by Rausch et al. supported a tailored approach, showing that the co-administration of azole antifungals (e.g., posaconazole, voriconazole, and isavuconazole) with dose-adjusted VEN (typically to 100 mg/day) did not compromise treatment efficacy or increase the incidence of FN [75]. Similarly, Agarwal et al. showed that posaconazole increases VEN exposure 7–8-fold, reinforcing the need for ≥75% dose reduction during co-administration, an approach shown to be safe and effective [76]. However, a more recent analysis by the Rausch et al. group showed that while neutropenia was prolonged, no significant difference in neutrophil recovery time was observed between patients receiving azoles and those on alternative prophylaxis [77]. What further emerged is that the rate of IFIs was comparable to those reported in patients undergoing intensive chemotherapy, supporting the use of AFP in this setting. The results support the practical use of azole prophylaxis alongside VEN, provided dose adjustments are made appropriately. This aligns with the most recent European Conference on Infections in Leukaemia (ECIL)-10 guidelines, which recommend mold-active prophylaxis for AML patients experiencing prolonged neutropenia and mandate VEN dose adjustment when combined with moderate or strong CYP3A4 inhibitors [78]. In our cohort, AFP was administered heterogeneously, with predominant use of fluconazole and limited use of mold-active azoles. This variability reflects the need to balance the risk of IFI against the potential for pharmacokinetic interactions and the importance of maintaining adequate VEN exposure. This approach mirrors early studies of VEN in AML, in which strong CYP3A4 inhibitors tended to be excluded due to the lack of pharmacokinetic data. In the VIALE-A trial, AFP was administered to 41% of patients, including 21% who received mold-active azoles, 15% treated with echinocandins, and 5% with amphotericin B [77]. Additionally, all individuals in a Phase II trial receiving VEN combined with a 10-day DEC regimen were given either a mold-active azole or echinocandin [79]. However, neither study reported the IFI incidence.

Although our findings are too limited to support a fully personalized, risk-adapted approach to AFP in patients receiving VEN-based regimens, the well-established benefit of maintaining treatment continuity for blast clearance underscores the importance of optimizing supportive care. Enhancing infection prevention strategies may, therefore, improve outcomes by minimizing infectious complications and reducing therapy interruptions. Data on R/R patients, often heavily pretreated or previously exposed to HMA, remain scarce, highlighting the need for further validation of AFP strategies, ideally through risk-adapted or score-based approaches. 

### 4.6. Study Limitations and Future Perspectives

The present study has three additional main limitations. First, its monocentric nature and small sample size limit generalizability. Second, we did not perform minimal residual disease (MRD) monitoring. MRD negativity has emerged as a key prognostic factor. In a study by Kristensen et al., responders who achieved MRD negativity had prolonged post-transplant remissions and unreached median OS [80]. Thirdly, the limited genomic analysis precluded the classification of patients according to the 2017 ELN risk stratification, as well as the more recently proposed 2024 ELN framework [81] and the VEN-specific prognostic risk score for R/R patients undergoing HMA-VEN therapy [57].

Taken together, our findings reinforce the role of HMA-VEN combinations as an effective, safe, and versatile option in R/R AML, in both transplant-eligible and -ineligible patients, independently from response timing. They highlight the need for adaptive treatment strategies tailored to patient fitness and support continued therapy in the absence of early remission, provided that toxicities remain manageable. Future research should integrate MRD-based response monitoring and prospective trials to optimize treatment duration and sequencing in this high-risk population.

## 5. Conclusions

In this prospective real-world study, the combination of HMA and venetoclax demonstrated encouraging efficacy and an acceptable safety profile in a cohort of patients with relapsed/refractory AML. Importantly, our findings are consistent with and expand upon the growing body of literature supporting this therapeutic approach. Notably, we found that the timing of remission, whether early or delayed, did not significantly impact clinical outcomes, provided that treatment was continued and adverse events were appropriately managed.

Future research should focus on optimizing the treatment duration, dose intensity, and sequencing of HMA-VEN regimens. In particular, prospective studies incorporating MRD-guided strategies and real-time molecular monitoring will be essential to fully exploit the potential of venetoclax-based therapies and improve long-term outcomes in this challenging patient population.

## Figures and Tables

**Figure 1 jcm-14-05586-f001:**
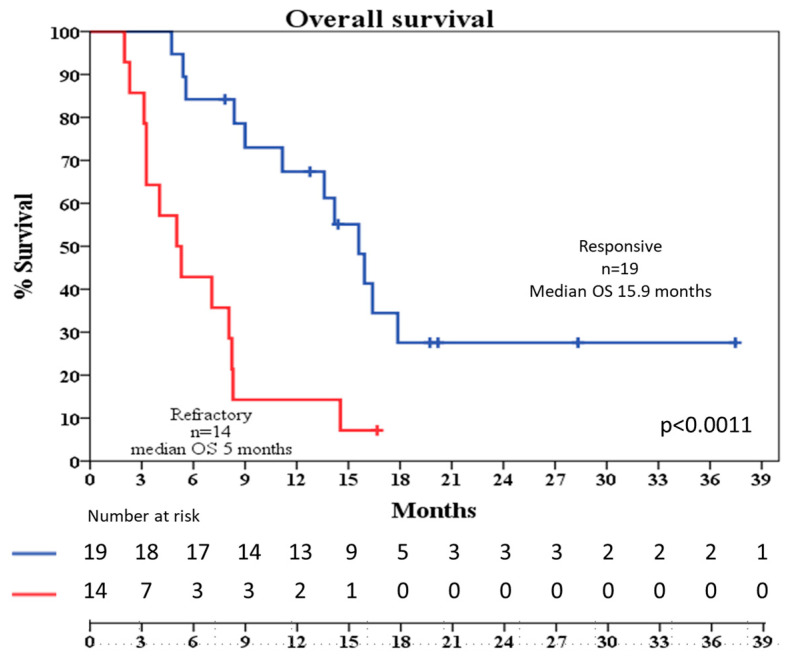
Median overall survival (OS) in months for responsive (blue) and refractory (red) patients. OS: overall survival; n: number of patients.

**Figure 2 jcm-14-05586-f002:**
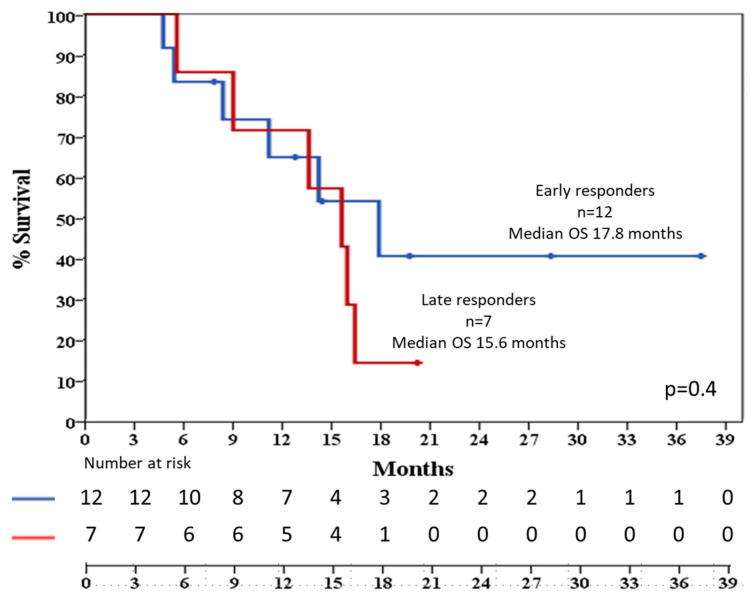
Median overall survival (OS) in months for patients showing early (blue) vs. late response (red). OS: overall survival; n: number of patients.

**Table 1 jcm-14-05586-t001:** Reasons for discontinuation of chemotherapy-based treatment stratified by disease status.

*Disease Status*	Reason for Discontinuation of Chemotherapy Program	N	% (Out of 33)
*Refractory*	No other treatment options	2	6%
	Uncontrolled infection	2	6%
	Patient choice	1	3.2%
	Chemoresistant	2	6%
*Relapsed—Early*	No other treatment options	9	27.3%
	Uncontrolled infection	4	12.1%
	Patient choice	2	6%
	Relapse after HSCT	2	6%
*Relapsed—Late*	No other treatment options	6	18.2%
	Uncontrolled infection	2	6%
	Patient choice	0	0%
	Relapse after HSCT	1	3.2%

Percentages are calculated on the total cohort (n = 33). Refractory patients are defined as those with no response to treatment before HMA/VEN; early and late relapse are categorized based on time from last remission to HMA/VEN start (if < or ≥12 months). HSCT: hematopoietic stem cell transplantation. N: number of patients.

**Table 2 jcm-14-05586-t002:** Baseline features and composite complete remission (cCR) rate according to clinical and molecular characteristics.

Parameter		Total Cohort (n/33)	cCR	*p*-Value
Age	<50 years old	7 (21.2%)	7/7 (100%)	0.01
	≥50 years old	26 (78.8%)	12/26 (46.2%)	
Gender	Male	15 (45.4%)	9/15 (60%)	0.8
	Female	18 (54.5%)	10/18 (55.6%)	
ECOG PS	ECOG PS 0–2	28 (84.8%)	19/28 (67.9%)	0.005
	ECOG PS 3–4	5 (15.2%)	0/5 (0%)	
t-AML	Yes	4 (12.1%)	3/4 (75%)	0.6
	No	29 (87.9%)	16/29 (55.2%)	
AML-MRC	Yes	10 (30.3%)	5/10 (50%)	0.7
	No	23 (69.7%)	14/23 (60.9%)	
Previous therapy	Chemotherapy	12 (36.4%)	6/12 (50%)	0.07
	Chemotherapy and AZA	7 (21.2%)	2/7 (28.6%)	
	HMA (AZA or DEC)	14 (42.4%)	11/14 (78.6%)	
Status from previous chemotherapy	Relapse	26 (78.8%)	13/26 (50%)	0.09
	Refractory	7 (21.2%)	6/7 (85.7%)	
Type of relapse *	Early relapse	17/26 (65.4%)	8/17 (47.1%)	0.7
	Late relapse	9/26 (34.6%)	5/9 (55.6%)	
Line of therapy	Second-line	26 (78.8%)	17/26 (65.4%)	0.08
	≥Third-line	7 (21.2%)	2/7 (28.6%)	
Combination	VEN + AZA	28 (84.8%)	18/28/ (64.3%)	0.065
	VEN + DEC	5 (15.2)	1/5 (20%)	
*FLT3*-ITD	Not detected	29 (87.9%)	18/29 (62.1%)	0.2
	Detected	4 (12.1%)	1/4 (25%)	
*NPM1*	w.t.	26 (78.8%)	16/26 (61.5%)	0.4
	Mutated	7 (21.2%)	3/7 (42.9%)	
Complex karyotype	No	28 (84.8%)	15/28 (53.6%)	0.3
	Yes	5 (15.2%)	4/5 (80%)	

AML-MRC: acute myeloid leukemia with myelodysplasia-related changes; AZA: azacitidine; cCR: composite complete remission; DEC: decitabine; ECOG: Eastern Cooperative Oncology Group performance status; HMA: hypomethylating agent; t-AML: therapy-related AML; VEN: venetoclax; w.t.: wild type; * (calculated on 26 relapsed patients).

**Table 3 jcm-14-05586-t003:** Clinical and biological characteristics at baseline between early and late responders.

Parameter		Early Responders N = 12	Late Responders N = 7	*p*-Value
Age	<50 years old	6 (50%)	1 (14.3%)	0.1
	≥50 years old	6 (50%)	6 (85.7%)	
Gender	Male	7 (58.3%)	3 (42.9%)	0.5
	Female	5 (41.7%)	4 (57.1%)	
ECOG PS	ECOG PS 0–2	12 (100%)	7 (100%)	NE
	ECOG PS 3–4	0	0	
Previous therapy	Chemotherapy	8 (66.7%)	3 (42.9%)	0.1
	Chemotherapy and AZA	2 (16.7%)	0	
	HMA (AZA or DEC)	2 (16.7%)	4 (57.1%)	
Status from previous chemotherapy	Relapse	8 (66.7%)	5 (71.4%)	0.8
	Refractory	4 (33.3%)	2 (28.6%)	
Type of relapse *	Early relapse	5 (62.5%)	3 (60%)	0.9
	Late relapse	3 (37.5%)	2 (40%)	
Line of therapy	Second-line	10 (83.3%)	7 (100%)	0.3
	≥Third-line	2 (16.7%)	0	
Combination	VEN + AZA	12 (100%)	6 (85.7%)	0.2
	VEN + DEC	0	1 (14.3%)	
*FLT3*-ITD	Not detected	11 (91.7%)	7 (100%)	0.4
	Detected	1 (8.3%)	0	
*NPM1*	w.t.	9 (75%)	7 (100%)	0.2
	Mutated	3 (25%)	0	
Complex karyotype	No	10 (83.3%)	5 (71.4%)	0.5
	Yes	2 (16.7%)	2 (28.6%)	

AZA: azacitidine; DEC: decitabine; ECOG: Eastern Cooperative Oncology Group performance status; HMA: hypomethylating agent; NE: not evaluable; VEN: venetoclax; w.t.: wild type; * (calculated on 26 relapsed patients).

## Data Availability

The raw data supporting the conclusion of this article will be made available by the authors without undue reservation.

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
