# Peer review of "Does the Timing of Response Impact the Outcome of Relapsed/Refractory Acute Myeloid Leukemia Treated with Venetoclax in Combination with Hypomethylating Agents? A Proof of Concept from a Monocentric Observational Study"

_jcm, 2025, doi:10.3390/jcm14155586_

Round 1

Reviewer 1 Report (Previous Reviewer 3)

Comments and Suggestions for Authors

The revised manuscript has addressed most of the previous concerns. However, there is one remaining issue. Line 300-315 contains an extra “Figure 1”. It looks like to be something displaced as the main text did not seem to talk about the OS in early vs late responders. The authors are expected to double check.

Author Response

Thank you for your suggestion.

We corrected the manuscript and double-checked it

Reviewer 2 Report (New Reviewer)

Comments and Suggestions for Authors

Thank you for the opportunity to read and review this insightful work. This is an excellent study concerning a current conundrum in the AML therapeutic field, and provides a positive outlook on treatment options. The authors have carefully considered several phenotypic aspects which may impact VEN-HMA therapeutic effectiveness, and have found such regimen to be beneficial in cases of R/R AML. Given the therapeutic inefficiency that currently exists in this field, the findings of the authors are very useful and will advance the field. 

Whilst it is clear the authors have considered some key mutational impacts, there should be a greater acknowledgement of the limitations of their genetic panel. They have referred to their use of the ELN 2017 guidelines, however the analysis completed does not take into account all recommended mutational screening outlined in the ELN 2017 Dohner et al. paper, that they have cited as reference 31. If it is achievable to further analyse the samples as per the rest of these genetic aberrations, this would significantly benefit the paper. If not now achievable due to time passed or resource limitations, a more detailed rationale as to the selection of NPM1 and FLT3, over other mutations that are part of the ELN recommendations such as TP53 and ASXL1, should be provided and explained more clearly. This includes perhaps in the limitations section of the discussion (line 397 onwards).

With regards to the figures, both Kaplan Meier plots are labelled as Figure 1. It would be beneficial to label these either as individual figures, or as Figure 1A and 1B. These should also be appropriately referred to in the text.

There are also a number of minor revisions that could be made to enhance this article further.

For example, on line 35 abstract, the full term for ECOG should be included prior to its abbreviation, as the authors have helpfully done with the rest of the abbreviations included in the abstract.

In the paragraph spanning lines 241-247, the structure is complex to follow. Perhaps the authors could consider abbreviation of this information into a table, or a summary of this information in this paragraph. 

In the paragraph spanning lines 337-340, the authors briefly discuss the need for treatment hold due to incidences of haematological toxicity, particularly with respect to the late responders. Expansion of this concept would be beneficial, including discussion of the length of time of treatment hold, and the point in treatment at which this occurred. 

To extend discussion further regarding the impacts on transplant bridging, the authors may consider including more details regarding the success of HSCT following VEN-HMA therapy. This could also include a greater explanation of the time frames during which the five patients discussed in line 371 remained in remission. 

Overall, this is a very good article that considerably advances this field, and I am happy to recommend this for publication, following editing in the aforementioned manners. 

Author Response

Reviewer 3 Report (New Reviewer)

Comments and Suggestions for Authors

The article titled "Does the Timing of Response Impact the Outcome of Relapsed/Refractory Acute Myeloid Leukemia Treated with Venetoclax in Combination with Hypomethylating Agents?" presents a single-center prospective study involving 33 adult patients with relapsed or refractory AML. It evaluates the efficacy, safety, and response dynamics to treatment with venetoclax combined with azacitidine or decitabine. This publication represents a significant contribution to the understanding of treatment for relapsed and refractory AML, confirming the effectiveness and safety of the HMA-VEN combination in everyday clinical practice. Moreover, it emphasizes that a delayed response to therapy should not be a reason for early treatment discontinuation. The article addresses a timely and clinically important issue, illustrating the therapeutic challenges associated with modern treatment regimens for R/R AML. The results are detailed and clear, encompassing both efficacy and safety outcomes, as well as analyses of different patient subgroups and factors affecting therapeutic response. However, I have a few suggestions:

  1. The discussion comes across as somewhat scattered. Given the complexity of the study, I recommend organizing the discussion into subsections, similar to the approach used in the results section.
  2. The expansions of abbreviations like BAK and BAX are missing.
  3. In the sentence in lines 119–122, it is not necessary to expand the abbreviations AZA and DEC.
  4. Please correct minor editorial errors such as double spaces.

Author Response

Reviewer 4 Report (New Reviewer)

Comments and Suggestions for Authors

Dear Editor and Authors,

I would like to thank you for the opportunity that you gave me to review the manuscript “Does the timing of response impact the outcome of relapsed/refractory acute myeloid leukemia treated with venetoclax in combination with hypomethylating agents?” (jcm-3776507). While the presented data are quite interesting, several issues, mainly regarding results presentation, should be addressed in order to make it worthy of publication. Please find below my comments and suggestions regarding your work.

Title:

Please include the type of study in the title.

Abstract:

-I think that before the beginning of the sentence “To assess…”, you should add the phrase “The aim of this study was to…”

Keywords:

-Please include more keywords while presenting them in alphabetical order.

Introduction:

-The introduction section is very well-written, providing some evidence regarding the rationale and the necessity of this study.

Materials and methods:

-In the 2.1 section, please include the approval code of your study from your local bioethics committee.

-Please describe the inclusion and exclusion criteria of your study using bullet points.

-How did you define persistent neutropenia?

-In the statistical analysis section, please explain how you assessed the distribution of normality of the variables.

-Did you also perform Cox regression analysis? If yes, please mention it in the Statistical analysis section.

Results:

-When mentioning FLA-Ida, please explain what agents were included in this regimen.

-Moreover, please include a Table with all the baseline characteristics and previous treatment lines of the patients (such as age, gender, previous treatments, AML [tAML, MDS progression], ECOG, genetic characteristics, risk group, previous allo-HSCT, laboratory values).

-Explain the HSCT abbreviation (also in the abstract). Be sure that you have explained all the abbreviations the first time used in the manuscript.

-In Table 1, what do you present? The reasons that the patients received HMA/V?

-In tables/ figures, explain all the keywords and present them in alphabetical order.

-What antifungal prophylaxis do your patients receive?

-FN abbreviation is also not explained.

-“The number of treatment cycles did not significantly differ between responders and non-responders (Mann–Whitney U test, p = 0.144).” Responders were those who achieved CR and PR?

-Line 267: “19/28 (67.9%) patients…” Never start a sentence with a number. Please write it as: Nineteen out of 28 (67.9%)…, be sure that your manuscript is consistent in this.

-In Table 2, you mention “Overall response rate”. You mean CR? Moreover, please use the abbreviation ORR even in the title.

-Did you perform univariate analysis regarding Response and the other studied factors? It would be beneficial.

-In line 278, “05%CI” should be “95% CI”.

-“Patients who achieved CR or CRi had significantly better survival (median OS: 15.9 months, 95%CI 12-19) compared to non-responders (median 280 OS: 5 months, 95%CI 2.7-7).” In which figure is this reported? Please mention it in the manuscript.

-The figure on page 9 should be listed as Figure 2. Additionally, please explain the abbreviations used in the figures.

-Line 325: “Of transplanted patients, 5 remained in remission at last follow-up.” What is the follow-up period?

-Line 325: “1 patient..” Please use: “One patient..”

-Please include more data regarding infectious complications in these patients. Did your patients also receive antibiotic and antiviral prophylaxis? How many developed bacterial, viral, and fungal (invasive fungal disease) infections? What kind of pathogen? Did infections have an impact on OS?

-Did the rate of febrile neutropenia and Grade 3/4 infection differ between early and late responders?

-Generally, I think that a Table comparing the characteristics (baseline and other) between early and late responders is essential. A univariate analysis between early response and several factors should be done.

Discussion

-The introduction of novel therapies, such as venetoclax, and of cellular therapies has been associated with the risk of invasive fungal diseases, and this should be highlighted in the discussion section (PMID: 40057633).

Round 2

Reviewer 4 Report (New Reviewer)

Comments and Suggestions for Authors

The authors have answered all of my comments and now the manuscript is ready for publication. I want to congratulate the authors for their high quality work.

Author Response

This manuscript is a resubmission of an earlier submission. The following is a list of the peer review reports and author responses from that submission.

Round 1

Reviewer 1 Report

Comments and Suggestions for Authors

This paper reports on treatment with Ven-HMA in R/R AML. The following revision points must be considered:

  • A previous Italian cohort study (Tidisco et al. 2023) to which the authors of the present study participated, is published: Is there a degree of overlap between the two studies? 
  • Characterization of refractory and relapsed disease must be refined: Percentage of refractory and relapsed cases, late (>12 months) and early relapses, refractory disease after intensive chemotherapy or not.
  • It is not clear if the patients had previously received HMA: If yes, in what percentage?
  • Pre-treatment hematologic values must be reported (WBC count, percentage of BM blasts).
  • The above characteristics should be reported in a Table, and response rate should be reported for each category (instead of number of patients per parameter among responders vs. non-responders reported in current Table 1).
  • Treatment characteristics: What was the median number of cycles of VEN-HMA, median time to response and median response duration? 

Reviewer 2 Report

Comments and Suggestions for Authors

The authors performed a single-center retrospective analysis in Italy to assess the efficacy and safety of VEN+AZA or VEN+DEC treatment in patients with relapsed or refractory AML. This study demonstrated the efficacy of these treatments even in second-line or third-line treatment. This study yields intriguing findings; nonetheless, the restricted sample size may hinder the identification of statistically significant differences in the analysis, necessitating caution in interpretation. Please revise the manuscript in accordance with the feedback given by the reviewer.

Major comments

Please delineate the criteria for the cessation and restart of treatment in response to adverse events linked to VEN.

The discussion of the causes for the substantial link between treatment efficacy and patient performance status is inadequate.

Despite the assertion that there was no statistically significant difference in the efficacy of VEN+AZA compared to VEN+DEC, this variation may be attributable to an insufficient sample size. Despite the assertion that there was no significant difference in response rates between second-line and third-line therapies, this variation may be attributable to the limited sample size and inadequate statistical power. Both of these signify significant constraints of the study, and the authors ought to incorporate a description of these limitations into the manuscript.

Latest research indicates that mitotic aberrations, significant to the mechanism of action of DEC, may arise for reasons beyond DNA demethylation.  The phenotypic study indicates that the mitotic aberrations caused by DEC are due not to DNA demethylation, but rather to bridge structures formed from abnormal covalent interactions between DNMT1 and DNA. As a result, DEC might work better when paired with ATR inhibitors (Ceralasertib) or CHK1 inhibitors (Rabusertib) than when combined with VEN.

Reference: Yabushita T, et al. Mitotic perturbation is a key mechanism of action for decitabine in myeloid tumor treatment. Cell Rep. 2023 Sep 26;42(9):113098.

Minor comments

Please furnish references for the Common Terminology Criteria for Adverse Events (CTCAE).

The results section lacks consistency in its numbering, with 3.3 succeeding 3.6.

Figure 1 should additionally incorporate the number at risk.

Reviewer 3 Report

Comments and Suggestions for Authors

The work from Longo et al. presents an retrospective study on the safety and efficacy of the treatment of venetoclax (VEN) combined with hypomethylating agents (HMAs) such as azacitidine (AZA) or decitabine (DEC) in a cohort of 33 adult patients with relapsed/refractory acute myeloid leukemia (R/R AML). They found a higher response in the VEN+AZA than VEN+DEC group, and that there is significant correlation between the response and the ECOG performance. They also report common hematologic toxicities and infrequent but manageable non-hematologic adverse effects in VEN-HMA treatments.

Comments and suggestions:

  1. Section 3.1 “Patient Characteristics”, it would be better if the authors could tabulate the age, the sex and previous treatment-related toxicity in addition to Table 1. This will be helpful to delineate the association of the safety and efficacy to covariates.
  2. Did the authors collect the methylation status of loci such as CEBPA, FLT3, and NPM1? It is curious to see how the benefit of HMAs correlates with the methylome aberration.
  3. Line 206, Figure 1, the authors are recommended to add the number of survivals in each time event at the bottom of the figure.
  4. The sample size is rather small (n=33), therefore the marginal significance in the difference between VEN+AZA vs VEN+DEC (Line 189, Table 1) may be due to the statistical power. The authors may acknowledge this limitation.

Round 2

Reviewer 1 Report

Comments and Suggestions for Authors

While I appreciate the authors's effort to revise the manuscript, the content constitutes overlapping publication that is not scientifically justified by any differential analysis, endpoint, or target audience. It should also be noted that local regulations (references 21 and 22) do not override established scientific standards for publication. 

Reviewer 2 Report

Comments and Suggestions for Authors

The authors have adequately addressed the reviewer's comments. The reviewer has no other comments and endorses the publication of the revised text.